# The impact of sonication cultures when the diagnosis of prosthetic joint infection is inconclusive

**Taiana Cunha Ribeiro[1], Emerson Kiyoshi Honda[2]◯, Daniel Daniachi[2]◯, Ricardo de Paula Leite Cury[2]◯, Cely Barreto da Silva[3], Giselle Burlamaqui Klautau[1], Mauro Jose Salles◯[1,4] ***

**1** Internal Medicine Department, Infectious Diseases Clinic, Santa Casa de São Paulo School of Medical Sciences, São Paulo, Brazil, **2** Orthopedic and Traumatology Department, Irmandade Santa Casa de Misericórdia de São Paulo, São Paulo, Brazil, **3** Department of Laboratory Medicine and Pathology, Irmandade Santa Casa de Misericórdia de São Paulo, São Paulo, Brazil, **4** Escola Paulista de Medicina, Universidade Federal de São Paulo, Division of Infection Diseases, São Paulo, Brazil

◯ These authors contributed equally to this work.

* salles.infecto@gmail.com

## Abstract

### Background

In the absence of a gold standard criterion for diagnosing prosthetic joint infections (PJI), sonication of the removed implant may provide superior microbiological identification to synovial fluid and peri-implant tissue cultures. The aim of this retrospective study was to assess the role of sonication culture compared to tissue cultures for diagnosing PJI, using different consensus and international guidelines for PJI definition.

### Methods

Data of 146 patients undergoing removal of hip or knee arthroplasties between 2010 and 2018 were retrospectively reviewed. The International Consensus Meeting (ICM-2018), Musculoskeletal Infection Society (MSIS), Infectious Diseases Society of America (IDSA), the European Bone and Joint Infection Society (EBJIS), and a modified clinical criterion, were used to compare the performance of microbiological tests. McNemar´s test and proportion comparison were employed to calculate *p*-value.

### Results

Overall, 56% (82/146) were diagnosed with PJI using the clinical criteria. Out of these cases, 57% (47/82) tested positive on tissue culture and 93% (76/82) on sonication culture. Applying this clinical criterion, the sensitivity of sonication fluid and tissue cultures was 92.7% (95% CI: 87.1%- 98.3%) and 57.3% (95% CI: 46.6%-68.0%) (p<0.001), respectively. When both methods were combined for diagnosis (sonication and tissue cultures) sensitivity reached 96.3% (95% CI: 91.5%-100%). Sonication culture and the combination of sonication with tissue cultures, showed higher sensitivity rates than tissue cultures alone for all diagnostic criteria (ICM-18, MSIS, IDSA and EBJIS) applied. Conversely, tissue culture provided greater specificity than sonication culture for all the criteria assessed, except for the

**Data Availability Statement:** All relevant data are within the manuscript and its Supporting information files.

**Funding:** The author(s) received no specific funding for this work.

**Competing interests:** The authors have declared that no competing interests exist.

EBJIS criteria, in which sonication and tissue cultures specificity was 100% and 95.3% (95% CI: 87.8–100%), respectively (p = 0.024).

## Conclusions

In a context where diagnostic criteria available have shortcomings and tissue cultures remain the gold standard, sonication cultures can aid PJI diagnosis, especially when diagnostic criteria are inconclusive due to some important missing data (joint puncture, histology).

## Introduction

Early accurate diagnosis of prosthetic joint infection (PJI) is made even more challenging in clinical practice by the lack of a gold standard test. In contrast, there have been several PJI diagnostic criteria suggested by consensus and international guidelines based on a composite of clinical signs and symptoms, blood and synovial fluid inflammatory biomarkers, histopathological abnormalities, and microbiological identification. The presence of sinus tract involving both prosthesis and skin or the identification of the same pathogen in two or more culture samples defines PJI for all of the currently proposed criteria (IDSA-Infectious Diseases Society of America, MSIS- Musculoskeletal Infection Society, ICM 2018-International Consensus Meeting and EBJIS-European Bone and Joint Infection Society) [1–5].

Traditionally, the first step in the diagnostic workup of a patient suspected of having an infected arthroplasty is testing for C-reactive protein (CRP), and erythrocyte sedimentation rate (ESR), but many clinical and microbiological aspects may impact the results and validity of these serum biomarkers [6]. The second step usually relies on the quantification of white-blood cells, percentage of granulocytes, and microbiological analysis of the synovial fluid through joint puncture [7]. Indeed, joint aspiration for synovial fluid inflammatory biomarkers and microbiological analysis has been regarded as one of the crucial pre-operative tests for the diagnosis of PJI [2, 8]. Unfortunately, this important diagnostic pre-operative step may not be carried out due to the low volume of joint fluid to be aspirated [9]. Furthermore, in the clinical practice the orthopaedic surgeon may be reluctant to performing joint aspiration due to the concern of joint and prosthesis contamination during the procedure [10]. Nevertheless, upon a clinical suspicion of PJI, synovial fluid analysis should always be attempted pre-operatively through joint aspiration [2].

Microbiological identification is vital for aiding conclusive diagnosis of PJI, particularly in cases where some of the assessments included in the criteria have not been carried out, in addition to ensure successful treatment outcomes [11–14]. However, some authors have shown that tissue and joint fluid cultures provide low sensitivity and high rates of false-negative results [15–18]. This poor performance might be related to the presence of low virulence microorganisms, previous antibiotic use, failure in the use of enriched culture media or insufficient sample incubation time [13, 19].

In this context, the sonication technique applied as an adjuvant method seems to optimize microbiological identification in cases of low-virulent biofilm-related microorganisms, thereby improving the diagnosis of chronic PJI [17, 20]. Both ICM 2018 and EBJIS endorsed sonication culture as an adjunct to tissue and synovial fluid cultures for diagnosing PJI [4, 5, 21]. In recent years, a number of authors have demonstrated that sonication fluid culture provides greater sensitivity than both tissue and synovial fluid cultures, reporting levels of 78–97% [17,

18, 22, 23]. In contrast, others published their results showing a stand-in against culturing son-
ication fluid of retrieved implants [24, 25]. Whether the role of sonication culture on the arma-
mentarium of PJI diagnosis remains under debate, many previous publications did not back-
up their findings on the recent consensus statements and international guidelines for PJI diag-
nosis, which may bias their results. On the other hand, the increment of sonication within the
methodological options for identifying the etiologic agent may allow PJI diagnosis to be estab-
lished even when some important variables are missing, such as those measuring pre-operative
synovial fluid biomarkers abnormalities.

Against this backdrop, the present study sought to assess the performance of sonication
fluid culture compared to periprosthetic tissue cultures for diagnosing PJI among patients
undergoing prosthetic hip or knee joint revision, using different consensus and international
guidelines for PJI definition (IDSA, ICM 2018, and EBJIS). Additionally, sonication was also
evaluated as an adds-on upon a subset of patients with incomplete (synovial fluid biomarkers
have not been assessed) diagnosis of PJI.

## Materials and methods

### Study design and setting

A retrospective observational study was carried out according to the Standards for Reporting
Diagnostic Accuracy (STARD) guideline, to assess the accuracy of sonication culture com-
pared to tissue cultures for diagnosing PJI, using different consensus and international guide-
lines for PJI definition [26]. Patients undergoing total or partial hip or knee prosthetic joint
revision due to any reason, whose removed implants were sent for sonication between Septem-
ber 2010 and December 2018 were analyzed. The study was carried out at the Department of
Orthopedics and Traumatology of a large tertiary academic hospital, comprising more than
1,000 beds, in São Paulo, Brazil. During the study period 2,216 primary hip and knee prosthe-
ses and 487 revisions were performed for any reason, at our institution. The clinical and surgi-
cal treatment decisions for PJI have traditionally been based upon the validated Zimmerli's
criteria, and involve a daily multidisciplinary musculoskeletal infection group analysis that
includes orthopaedic surgeons, infectious disease physicians, and microbiologists [27].

Patients with fewer than 2 tissues samples sent for culture, whose implant was not sent for
sonication in an appropriate sterile plastic container, or subject to contamination during
implant removal, transportation, or laboratory culture processing, were excluded from the
study. The present study was approved by the Research Ethics Committee of the institution,
with permission granted prior to commencement (permit 2.195.577, 01/08/2017).

Patient demographics and comorbidities, arthroplasty site, previous orthopedic surgical
procedures, reported clinical signs and symptoms, number of tissue samples collected per
patient, histological abnormalities, time elapsed between prosthesis implantation and removal,
previous use of antibiotics in the 14 days leading up to arthroplasty removal and microbiolog-
ical identification in cultures were recorded.

### Diagnosis of prosthetic joint infection (PJI)

Our Institutional musculoskeletal infection team employed, up to July 2018 the MSIS diagnos-
tic criteria for PJI and switched to the ICM-2018 thereafter. For the purposes of this study, in
the absence of a gold standard criterion for diagnosing PJI, the following diagnostic criteria
were used to compare the performance of sonication fluid culture versus periprosthetic tissues
cultures: ICM-2018, IDSA, and the EBJIS [1–5]. For the remaining assessments, definitive
diagnosis of PJI was established using the modified clinical criteria published in previous stud-
ies [1, 15, 18]. This criterion includes presence of sinus tract, visualization of periprosthetic

purulent secretion, and histology disclosing acute inflammatory process (at least five neutrophils in each of five high-power fields, at ×400 magnification), thereby precluding the need for microbiological results. The IDSA, ICM-2018 and EBJIS comprises a combination of clinical, histological and microbiological results and consider the presence of either sinus tract or two positive cultures with the same pathogen as conclusive diagnosis of PJI [1–5]. Both ICM-2018 and EBJIS criteria recommend synovial fluid aspiration, and microbiological identification by sonication is provided only by EBJIS (S1 Fig) [2, 4, 5].

## Specimen collection and microbiological methods

During the surgical procedure, at least 4 samples of periprosthetic and bone tissue were collected aseptically, placed in duly labelled sterile containers, and sent to the microbiology and histopathology laboratory. The flow protocols for synovial fluid and tissue sample collection, transportation and processing were well established in the study institution and validated by previous publications [23, 28]. At the laboratory, tissue samples were homogenized in 3 ml of brain heart infusion (BHI) agar for 1 minute and inoculated onto aerobic blood agar, chocolate agar and anaerobic blood agar plates jar at 35˚C, and also in thioglycollate medium (BD Diagnostic Systems, Sparks, MD). The blood agar and chocolate agar plates were then incubated at 35–37˚ C for 6 days for aerobic and 14 days for anaerobic cultures. The thioglycollate broth was incubated for 14 days and in the event of bacterial growth (turbidity), the liquid was seeded onto blood agar plates (aerobic and anaerobic cultures). Microbiological methods for synovial fluids were similar, inoculating 0.1 mL onto agar plates and liquid broth and assessing aerobically and anaerobically. Colonies of isolated bacteria were subjected to Gram staining and phenotypic identification, including motility tests and manual biochemical tests such as those for catalase and coagulase (using rabbit plasma) for Gram-positive bacteria and fermentation of sugars and amino acids by Gram-negative bacteria, among others. The sensitivity profile was determined for all strains identified according to the prevailing CLSI (*Clinical Laboratory Standards Institute*) standards (Standardization of Antimicrobial Disk Diffusion Susceptibility Testing: Approved Standard– 8th Edition, 2010, Vol. 23 No 1).

## Arthroplasty sonication

The surgically removed arthroplasties were ideally submitted to sonication within 2 hours. The protocol for implant removal, transportation to the microbiology laboratory and carrying out of sonication and cultures of the sonicated fluid was standardized and validated as per previous publications [23, 28]. The removed prostheses were placed in hermetically sealed sterile polyethylene containers together with 50 to 250 mL of Ringer Lactate (depending upon the implant width) and then transported in plastic bags to prevent leakage contamination. Upon arrival at the laboratory, the containers holding implants were agitated vigorously by vortex for 30 seconds using a Vortex-Genie 2 device (Scientific Industries, Inc., Bohemia, NY, USA). The containers were then sonicated in an ultrasound bath (BactoSonic; Bandelin GmbH, Berlin, Germany) for 5 minutes at a low frequency (40 ± 2 kHz) and high density of 0.22 ± 0.04 W/cm2, followed by agitation for a further 30 seconds in a vortex [23]. The sonicated fluid (50 to 250 ml) was divided into sterile tubes and centrifuged for 5 minutes. The supernatant was aspirated, leaving 0.5 ml (100-fold concentration), and aliquots of 0.1 ml of concentrated sonicate fluid were then plated onto aerobic sheep blood, chocolate, and anaerobic sheep blood agar, and incubated aerobically at 37˚C for 7 days and anaerobically at 37˚C for 14 days and inspected daily for bacterial growth. Additionally, 4 ml of the remaining concentrated sonication fluid was also inoculated in 10 ml of thioglycolate broth, plated as described above, and incubated aerobically at 35˚C to 37˚C in 5% CO2 for 2 days, and anaerobically at 37˚C for 14

days. In the event growth was detected, the number of colonies forming units (CFU) for each morphology was recorded. Due to the addition of a concentrating step to the sonication fluid culture, a density $\geq$ 50 CFU/plate of sonicated fluid is considered significant and used for ideal sensitivity and specificity analyses [23, 28]. All plates exhibiting positive growth were quantified and identified according to the routine established by the laboratory, based on the morphology and staining property visualized on Gram staining. Low virulence microorganisms (*Staphylococci epidermidis*, *Corynebacterium* spp., *Chryseobacterium* spp., *Bacillus* spp. and *Micrococcus* spp.) were considered pathogenic when the organism was found in at least 2 different culture samples [1–3].

Retrieved implant cases due to aseptic loosening were used for negative controls and equally processed as described for the retrieved infected arthroplasties.

## Statistical analysis

The demographic characteristics of patients were expressed in frequencies and percentages or means and standard deviations (SD). Sonication culture, tissue culture and the combined methods were evaluated with sensitivity, specificity, positive predictive value, negative predictive value, and accuracy, for all diagnostic criteria used. In order to identify the impact of the total number of tissue samples collected per patient in the sensitivity rates, three subgroups were created and analyzed: 1: between 2 to 4 tissue samples; 2: between 5 to 7 tissue samples; 3: at least 8 tissue samples collected. Culture sensitivity were also explored among patients with the modified clinical criteria according to time span between index surgery and the explantation; clinical and pathological abnormalities (sinus tract, visible purulence during surgery, positive histology); previous use of antibiotics; and virulence of microorganism identified. Sensitivity and specificity of tissue and sonicated fluid cultures were compared using McNemar's test and comparison of proportions was employed to calculate *p*-value. Differences with a p-value $\leq$ 0.05 for a 95% Confidence Interval (95% CI) were considered statistically significant. All data were analyzed using the SPSS statistical software package for Windows, version 13.0 (IBM Corporation, Chicago, IL).

## Results

### Study population and devices

Overall, 146 patients undergoing revision of prosthetic hip or knee joints for any reason and whose removed implant was submitted to sonication and tissue cultures, were included in the study. In the overall sample, median age was 66 years (range 17–96 years) and 58.2% of patients were female. Of the total prostheses revised, 71% were hip joints, whereas only 29% were knee joints. Most of the arthroplasties revised (76%) were primary prostheses. Demographics, clinical characteristics, and number of patients diagnosed with PJI according to different criteria (clinical, IDSA, ICM and EBJIS) of the study population are shown on Table 1. Of total patients assessed, 56% (82/146) were diagnosed with PJI using the clinical criteria, 57% (83/146) the IDSA, 53% (77/146) the ICM and 71% (103/146) by the EBJIS criteria.

### Accuracy of sonication fluid and tissue cultures according to different definitions of PJI

The assessment of all patients revealed that 39% (57/146) tested positive on tissue samples for at least one microorganism whereas 67% (98/146) had microbiological detection using sonication fluid. Only 29% (43/146) of patients had negative cultures when both methods were used.

**Table 1. Demographics, clinical characteristics and number of patients diagnosed with PJI according to different criteria (clinical, IDSA, ICM and EBJIS) among 146 patients undergoing prosthetic joint revision.**

| Demographics[a] | Number of Patients N (%) Total = 146 |
|---|---|
| • Age (median [range]) (years) | 66 (17–96) |
| • Female sex (no. [%]) | 85 (58%) |
| **Clinical characteristics** (no. [%]) | |
| • Diabetes mellitus | 40 (27%) |
| • Rheumatoid arthritis | 27 (18%) |
| • Tobacco use | 20 (14%) |
| • Coronary heart disease | 11 (8%) |
| • Solid organ neoplasm | 10 (8%) |
| • Steroid use | 10 (8%) |
| • Heart failure | 10 (8%) |
| • Chronic kidney disease | 5 (4%) |
| • Alcohol abuse | 5 (4%) |
| **Arthroplasty Site** (no. [%]) | |
| • Hip | 103 (71%) |
| • Knee | 43 (29%) |
| **Arthroplasty Type** (no. [%]) | |
| • Primary | 111 (76%) |
| • Revision | 35 (24%) |
| **Time since prosthetic implantation** (no. [%]) | |
| • < 3 months | 21 (14%) |
| • 3–24 months | 33 (23%) |
| • > 24 months | 92 (63%) |
| **Signs and symptoms of PJI** (no. [%]) | |
| • Pain | 145 (99%) |
| • Purulent secretion around prosthesis | 66 (45%) |
| • Hyperemia | 43 (29%) |
| • Presence of sinus tract | 21 (14%) |
| • Prosthesis dislocation | 18 (12%) |
| • Fever | 5 (3%) |
| **ESR** | |
| • No. of patients with the data (%) | 125 (86%) |
| • MEAN | 44.09 mm/h |
| **CRP** | |
| • No. of patients with the data | 142 (97%) |
| • MEAN | 6.67 mg/dL |
| **Patients diagnosed with PJI** (no. [%]) | |
| • Clinical Criteria | 82 (56%) |
| • IDSA Criteria | 83 (57%) |
| • ICM Criteria | 77 (53%) |
| • EBJIS Criteria | 103 (71%) |

[a] All percentages are in relation to the number of subjects with osteosynthesis-associated infection (OAI) or noninfected osteosynthesis (NIO), unless otherwise indicated. PJI: Prosthetic Joint Infection; ESR: erythrocyte sedimentation rate; CRP: C-reactive protein; ICM: *International Consensus Meeting*; IDSA: *Infectious Diseases Society of America*; EBJIS: *European Bone and Joint Infection Society*.

**Table 2. Description of PJI cases according to different criteria, sensitivity and specificity of tissue and sonication cultures in 146 patients undergoing prosthetic joint revision.**

| PJI criteria | Total number of PJI n (%) | Sensitivity, % (95% CI) | | p-value tissue vs. sonication | Specificity, % (95% CI) | | p-value tissue vs. sonication |
|---|---|---|---|---|---|---|---|
| | | Tissue | Sonication | | Tissue | Sonication | |
| **Clinical** | 82 (56%) | 57.3 (46.6–68.0) | 92.7 (87.1–98.3) | < 0.001 | 84.4 (75.5–93.3) | 65.5 (54.0–77.2) | 0.024 |
| **ICM** | 77 (53%) | 68.8 (58.5–79.1) | 94.8 (89.0–100.0) | < 0.001 | 94.2 (87.8–100.0) | 63.8 (52.5–75.1) | < 0.001 |
| **IDSA** | 83 (57%) | 65.1 (54.8–75.4) | 94.0 (88.0–100.0) | < 0.001 | 95.2 (89.0–100.0) | 68.3 (56.7–79.7) | < 0.001 |
| **EBJIS** | 103 (71%) | 53.4 (43.8–63.0) | 95.1 (90.9–99.3) | < 0.001 | 95.3 (87.8–100.0) | 100 (100–100) | 0.024 |

PJI: prosthetic joint infection; ICM: International Consensus Meeting; IDSA: Infectious Diseases Society of America; EBJIS: European Bone and Joint Infection Society. CI: 95% confidence interval, $p < 0.05$, McNemar's Test.

When assessing only PJI cases, the percentage of patients with positive cultures increased, with rates varying depending on the PJI criterion employed. Interestingly, positivity was higher in sonication fluid cultures than in tissue cultures for all of the diagnostic criteria applied in the present study. Data for number of patients diagnosed with PJI by the different criteria, together with sensitivity and specificity of the tissue and sonication cultures, are shown in Table 3. Using the ICM-2018 definition, sensitivity of sonication and tissue cultures were 94.8% (73/77) (95% CI: 89–100%) and 68.8% (53/77) (95% CI: 58.5–79.1%), (p<0.001) respectively. Using IDSA guidelines, sensitivity of sonication and tissue cultures were 94% (78/83) (95% CI: 88–100%) and 65.1% (54/83) (95% CI: 54.8–75.4%), (p<0.001) respectively. Sonication culture also showed greater sensitivity than tissue cultures for all the other diagnostic criteria applied, with the highest rate of 95.1% (98/103) (95% CI: 90.9%- 99.3%) being found for the EBJIS criteria (p<0.001; Table 2).

Sensitivity using the clinical modified criteria (presence of sinus tract, or visible purulent secretion, or positive histology for infection) was higher for sonication fluid cultures—92.7% (95% CI: 87.1–98.3%) than for tissue cultures—57.3% (95% CI: 46.6–68.0%), (p<0.001). Assessment of the test's specificity using the clinical modified criteria revealed that tissue culture had higher specificity than sonication culture—84.4% (95% CI: 75.5%- 93.3%) versus 65.5% (95% CI: 54.0%-77.2%), respectively (p = 0.024).

Specificity was significantly higher for tissue culture than sonication when using ICM and IDSA definitions. Conversely, only the EBJIS criterion had higher specificity for sonication than tissue cultures—100% and 95.3% (95% CI: 87.8–100%), respectively (p = 0.024; Table 2). Sensitivity, specificity, positive predictive value, negative predictive value and accuracy of tissue and sonication cultures used alone and in conjunction, for each diagnostic criterion proposed are shown in Table 3. Combining the two methods, tissue culture plus sonication culture provided greater sensitivity than tissue or sonication culture alone, irrespective of the criterion used (Table 3).

When modelling the effect of diverse numbers of tissue samples collected per patient, the sensitivity of tissue cultures varied according to the number of samples obtained intra-operatively. Fig 1 shows that collection of 5 or more tissue samples considerably improved test sensitivity. When sonication culture is used in conjunction with tissue cultures, sensitivity is also increased and remains unchanged irrespective of the number of samples collected during the surgical procedure (Fig 1).

**Table 3. Sensitivity, specificity, positive predictive value, negative predictive value and accuracy of tissue and sonication cultures isolated and of tissue and sonication combined, according to proposed diagnostic criteria in 146 patients undergoing prosthetic joint revision.**

| Clinical Criterion | Tissue | 95% CI | Sonication | 95% CI | Tissue and Sonication | 95% CI |
|---|---|---|---|---|---|---|
| Sensitivity | 57.3% | (46.6–68.0) | 92.7% | (87.1–98.3) | 96.3% | (91.5–100) |
| Specificity | 84.4% | (75.5–93.3) | 65.6% | (54.0–77.2) | 62.5% | (50.6–74.4) |
| PPV | 82.5% | (75.0–90.0) | 77.6% | (69.3–85.9) | 76.7% | (68.5–84.9) |
| NPV | 60.7% | (46.9–74.5) | 87.5% | (78.1–96.9) | 93.0% | (84.0–100) |
| Accuracy | 69.2% | (61.7–76.7) | 80.8% | (74.4–87.2) | 81.5% | (75.2–87.8) |
| **ICM Criterion** | **Tissue** | **95% CI** | **Sonication** | **95% CI** | **Tissue and Sonication** | **95% CI** |
| Sensitivity | 68.8% | (58.5–79.1) | 94.8% | (89.0–100.0) | 98.7% | (95.7–100) |
| Specificity | 94.2% | (87.8–100.0) | 63.8% | (52.5–75.1) | 60.9% | (49.4–72.4) |
| PPV | 93.0% | (85.2–100.0) | 74.5% | (65.9–83.1) | 73.8% | (65.3–82.3) |
| NPV | 73.0% | (63.8–82.2) | 91.7% | (82.5–100.0) | 97.7% | (92.4–100) |
| Accuracy | 80.8% | (74.4–87.2) | 80.1% | (73.6–86.6) | 80.8% | (74.4–87.2) |
| **IDSA Criterion** | **Tissue** | **95% CI** | **Sonication** | **95% CI** | **Tissue and Sonication** | **95% CI** |
| Sensitivity | 65.1% | (54.8–75.4) | 94.0% | (88.0–100.0) | 97.6% | (93.8–100) |
| Specificity | 95.2% | (89.0–100.0) | 68.3% | (56.7–79.7) | 65.1% | (53.3–76.9) |
| PPV | 94.7% | (87.9–100.0) | 79.6% | (71.6–87.6) | 78.6% | (70.7–86.5) |
| NPV | 67.4% | (57.7–77.1) | 89.6% | (79.4–99.8) | 95.3% | (87.8–100) |
| Accuracy | 78.1% | (71.4–84.8) | 82.9% | (76.8–89.0) | 83.6% | (77.6–89.6) |
| **EBJIS Criterion** | **Tissue** | **95% CI** | **Sonication** | **95% CI** | **Tissue and Sonication** | **95% CI** |
| Sensitivity | 53.4% | (43.8–63.0) | 95.1% | (90.9–99.3) | 98.1% | (95.0–100) |
| Specificity | 95.3% | (87.8–100.0) | 100.0% | (100.0–100.0) | 95.3% | (87.8–100) |
| PPV | 96.5% | (90.9–100.0) | 100.0% | (100.0–100.0) | 98.1% | (95.0–100) |
| NPV | 46.1% | (35.7–56.5) | 89.6% | (79.4–99.8) | 95.3% | (87.8–100) |
| Accuracy | 65.8% | (58.1–73.5) | 96.6% | (93.2–100.0) | 97.3% | (94.3–100) |

PPV: positive predictive value, NPV: negative predictive value; ICM: International Consensus Meeting; IDSA: Infectious Diseases Society of America; EBJIS: European Bone and Joint Infection Society. CI: confidence interval.

## Subgroup analyses

The sensitivity of the tissue and sonication fluid samples according to clinical and microbiological characteristics, applying the PJI clinical criteria, are shown in Table 4. In the subgroup analysis, the smaller numbers may limit power, but the sensitivity of sonication was higher than tissue culture and showed statistical significance for most subgroup studied (late PJI, patients with visible purulence, positive histology, previous antibiotic use, and for virulent and low-virulent microorganisms). However, for patients with early and delayed PJI and the presence of sinus tract, no statistical significance was observed (Table 4).

## Microbiological assessment

The culture of most patients revealed only one etiological agent. Between 23–29% of cultures exhibited polymicrobial flora in tissue samples compared to 21–26% in sonication fluid samples, depending on the diagnostic criteria used.

The pathogen most commonly found in prosthetic infections was *Staphylococcus epidermidis*, being isolated in 22% of tissue cultures, but in 30% of sonication cultures, followed by *Staphylococcus aureus*, found in 13% and 17% of tissue and sonication cultures, respectively. Gram-negative bacilli were detected in 23% of tissue cultures and 36% of sonication cultures in patients diagnosed with PJI using the clinical criteria (S1 Table). Low-virulence microorganisms

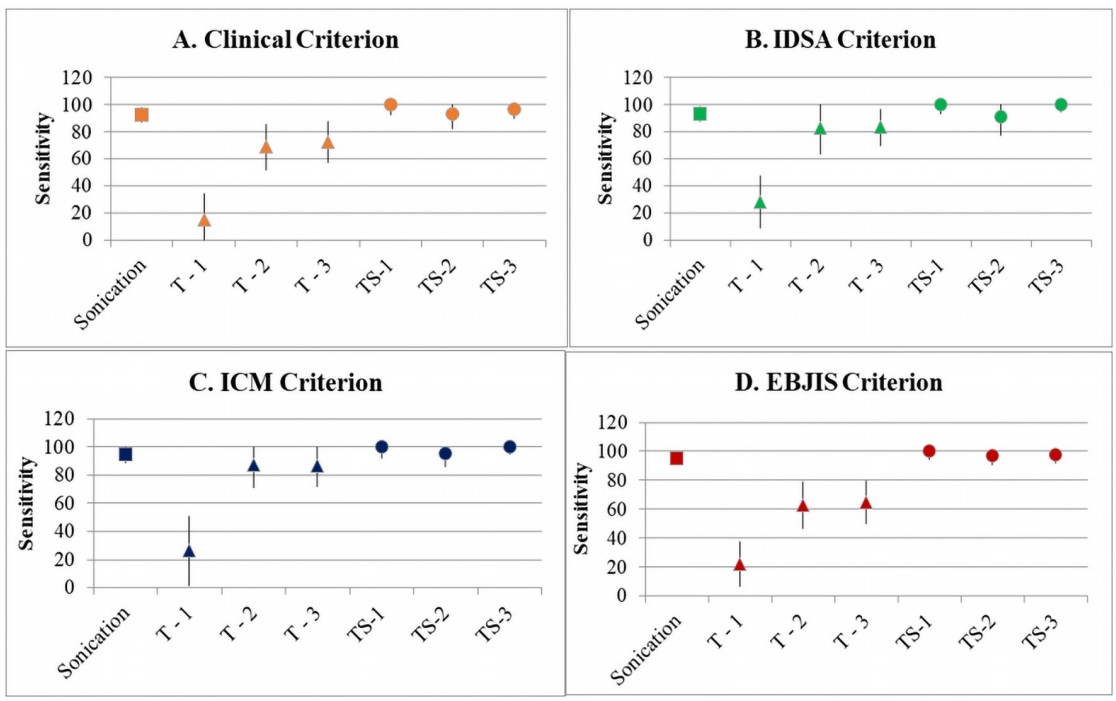

**Fig 1. Sensitivity of sonication (S), tissue cultures (T) and tissue plus sonication (TS) for the following diagnostic criteria and according to number of tissue samples collected.** Diagnostic criteria: clinical modified (A), IDSA (B), ICM (C) and EBJIS (D). Number of tissue samples collected: 1: 2–4 tissue samples; 2: 5–7 tissue samples; 3: ≥8 tissue samples. PJI: prosthetic joint infection; ICM: International Consensus Meeting; IDSA: Infectious Diseases Society of America; EBJIS: European Bone and Joint Infection Society.

(*Staphylococcus epidermidis, Stenotrophomonas maltophilia, Corynebacterium* spp, *Cryseobacterium indologenes, Peptostreptococcus* spp and *Micrococcus*) were detected in 24% of tissue cultures and 38% of sonication cultures in PJI using the clinical criteria. Discordant and discrepant cultures are shown in S2 and S3 Tables, respectively, of the Supporting information section.

**Table 4. Assessment of clinical and microbiological characteristics and sensitivity of tissue and sonication fluid cultures in patients clinically diagnosed with PJI.**

| Subgroups of 82 patients with PJI according to clinical criteria | No. (%) of patients | Sensitivity, % (95% CI) | | p-value |
|---|---|---|---|---|
| | | **Tissue** | **Sonication** | |
| **Time since prosthetic placement** | | | | |
| < 3 months | 15 (18%) | 60.0 (35.2–84.8) | 93.3 (77.1–100.0) | 0.085 |
| 3–24 months | 28 (34%) | 64.3 (46.6–82.0) | 89.3 (75.4–100.0) | 0.058 |
| > 24 months | 39 (48%) | 51.3 (35.6–67.0) | 94.9 (86.7–100.0) | < 0.001 |
| **Clinical abnormalities** | | | | |
| Sinus tract | 21 (26%) | 66.7 (46.5–86.9) | 90.5 (75.0–100.0) | 0.133 |
| Visible purulent secretion | 66 (80%) | 59.1 (47.2–71.0) | 92.4 (86.0–98.8) | < 0.001 |
| Positive histology | 60 (73%) | 50.0 (37.3–62.7) | 95.0 (88.5–100.0) | < 0.001 |
| **Previous antimicrobial use[a]** | | | | |
| Yes | 19 (23%) | 42.1 (19.9–64.3) | 84.2 (63.7–100.0) | 0.019 |
| **Virulence of microorganism identified** | | | | |
| Virulent microorganisms | 53 (65%) | 64.2 (51.3–77.1) | 100.0 (97.1–100.0) | < 0.001 |
| Low-virulence microorganisms | 26 (32%) | 50.0 (30.8–69.2) | 88.5 (73.6–100.0) | 0.007 |

[a] Previous antimicrobial use in the fifteen days prior to prosthesis revision surgery. CI: 95% confidence interval, $p < 0.05$.

## Discussion

In routine clinical practice, it is well known that all adjunctive laboratory tests recommended by consensus and guidelines to fulfill the diagnostic criteria of PJI may not be always carried out. For instance, quantitative analysis of total leukocytes and percentage neutrophils in synovial fluid, although included in most PJI diagnostic criteria (ICM, EBJIS), are often skipped. Therefore, a "clinical criterion" independent of microbiological factors was also adopted. Interestingly, irrespective of the type of criteria elected for diagnosing PJI (clinical modified, ICM, IDSA or EBJIS), the use of the sonication fluid culture proved superior to traditional tissue culture methods in our studied patients. Moreover, the diagnosis of PJI was improved by using sonication fluid cultures for all the criteria used in clinical practice. Despite these findings, periprosthetic tissue cultures remain highly specific and the gold standard for microbiological diagnosis [24, 25, 29].

Our study has limitations, including the inherent drawbacks of retrospective single-center studies, where data were collected from databases often with incomplete records. Not all elements of the diagnostic criteria for JPI were available. Data on CRP and ESR tests were available for a high percentage of patients, while only five patients underwent preoperative synovial fluid collection. Furthermore, tissue specimens were not sent for histological analysis in 23% (34/146) of patients. Thus, for most patients with confirmed PJI, diagnosis was reached postoperatively based on culture results, significantly delaying the appropriate antimicrobial treatment.

The present study meant to aid readers to answer questions regarding the accuracy of sonication versus tissue cultures and the combination of both methods against currently diagnostic criteria (ICM-2018, MSIS, EBJIS), and when clinical suspicious is supported by few signs or tests (sinus tract or visible purulent or positive histology), which we named "modified clinical criterion". Some authors hold that tissue cultures offer better performance than sonication fluid culture, with sensitivity of 68–96% for tissue cultures versus 47–70% for sonication cultures reported [24, 25, 30]. They claim that sonication technique offers little benefit in the diagnosis of PJI when tissue cultures are performed using adequate standard methods. However, empirical antibiotic therapy is commonly used for early acute PJIs (postoperative and haematogenous) in clinical practice, a factor known to reduce the positivity of tissue cultures relative to sonication techniques. This reduction is likely due to the greater susceptibility of planktonic bacteria to antimicrobials compared to the bacteria found in biofilm [18, 25].

By contrast, other authors have demonstrated that sonication fluid culture provides greater sensitivity than both tissue and synovial fluid cultures [17, 18, 22, 23], possibly enhancing microbiological detection in biofilm-related orthopedic infections. Trampuz et al., were the first authors back in 2007 to found greater sensitivity for sonication cultures than for tissue cultures, and a non-statistically significant difference compared with synovial fluid cultures. The study also showed that collecting a higher number of tissue samples raised sensitivity of these cultures, with values ranging from 50% for only 2 tissues samples to 72.7% for ≥5 samples [18]. The findings of the present study corroborated these results, showing greater sensitivity for sonication fluid cultures than for tissue cultures, irrespective of the diagnostic criteria employed. Besides, tissue culture sensitivity was found to increase when ≥5 periprosthetic tissue samples were collected. The low sensitivity of tissue cultures found might be due to the inoculation of samples into thioglycollate medium or onto agar plates, since the use of blood bottle cultures for this purpose is not a standard procedure in our laboratory. The collection of at least five periprosthetic tissue specimens, the immediate dispatch of samples to the laboratory, sample processing in sterile pearl glass flasks for maceration and subsequent inoculation in automated blood culture bottles (BACTEC) have been shown to favor greater accuracy of

tissue cultures, as reported by numerous authors [15, 31–33]. However, we argue that these techniques of processing tissue samples are not routinely carried out at the hospital where the study was conducted, which may have negatively impacted the accuracy of these cultures.

Use of sonication technique together with tissue culturing in the investigation of PJI promoted better microbiological identification, with greater culture sensitivity even when few tissue samples were collected. These data confirm that sonication aids microbiological diagnosis of PJIs, when only a small amount of viable tissue is available for microbiological analysis.

In addition, sonication cultures yield better results in patients with previous antibiotic intake and in late infections (i.e. those occurring 24 months after implant placement) compared to early infections, as shown by more recent studies [20, 29]. This likely occurs because there is a greater number of bacteria at the bone-implant interface early in the infectious process, favoring periprosthetic tissue cultures. In later infections, however, the greatest bacterial inoculation is found within biofilm. Thus, the use of sonication promotes detachment of biofilm from the implant, thereby facilitating microbiological detection and improving culture sensitivity [25, 34]. The effects of sonication applied at low frequency and high intensity on biofilms have been proved to increase the accuracy of bacterial counts within cultures by the mechanical destruction of the biofilm extracellular matrix due to the effect of ultrasonic cavitation [35, 37]. Most importantly, the natural process of biofilm passive dispersion in which cell escape from the inner biofilm structure to return to its previous single-cell planktonic mode of growth is increased by sonication. The process of boosting biofilm dispersion by sonication is most likely responsible for improving microbiological diagnostic yield [35–37].

The assessment of the different diagnostic criteria proposed as a gold standard for diagnosing JPI revealed greater accuracy when using sonication culture. However, the IDSA, ICM and EBJIS criteria stipulate tissue cultures in their definition, while the EBJIS criterion also includes sonication culture. This limitation represents a major bias in accurate analysis of these cultures. Hence, only the clinical criterion, which encompasses the presence of sinus tract, visible purulent secretion, and consistent histopathological abnormality, provides optimal assessment of these microbiological methodologies.

Despite the high sensitivity of sonicated fluid cultures for all the criteria investigated, this method has low specificity, suggesting the technique may overestimate PJI diagnoses given the greater likelihood of false positives. In a recent publication, Dudareva et al. [24] (2018) questioned whether the microorganism isolated in sonication indeed causes an inflammatory or infectious process at the site or is merely inert within the biofilm, and emphasized that tissue sample culture is more sensitive and specific than sonication for the microbiological diagnosis of PJI. Additionally, the study by Grosso et al. [30] (2018) points out other limitations in the use of the sonication technique, such as having the necessary infrastructure for transporting the removed implant to the microbiology laboratory, the need for sterile containers, which should be relatively large to accommodate the implant, and exercising the required care in shipping and handling the prosthesis so as to minimize contamination [30].

With regard to the microbiology, the rates of pathogens found in the present study were consistent with data reported in the literature. Previous studies have shown greater involvement of Gram-positive bacteria in the hip and knee PJI, with *Staphylococcus epidermidis* and *Staphylococcus aureus* being the main pathogens [17, 38–40]. Chronic infections are associated with greater involvement of low virulence organisms, such as *S. epidermidis*, whereas early and acute infections are more commonly associated with *Staphylococcus aureus* and Gram-negative bacteria [39, 41]. Given that the present study included predominantly cases of PJI more than 3 months after implantation, there was a greater prevalence of *S. epidermidis* detected in tissue and sonication cultures.

In summary, the present study results suggest that, in a context where the diagnostic criteria available have shortcomings and tissue cultures remain the gold standard, the sonication technique can aid PJI diagnosis. The technique is especially useful when preoperative joint puncture has not been done, the periprosthetic tissue specimen available for collection is small, and the sample processing method is not ideal.

## Supporting information

**S1 Fig. Diagnostic criteria for Prosthetic Joint Infection (PJI) according to different guidelines, consensus and by a clinical criterion.** ICM: International Consensus Meeting; IDSA: Infectious Diseases Society of America; EBJIS: European Bone and Joint Infection Society. CRP: C-reactive protein; ESR: erythrocyte sedimentation rate.
(TIF)

**S1 Table. Distribution of microorganisms detected by sonication fluid and tissue cultures among 103 patients with microbiological identification and in 82 patients with clinically PJI.** [a] Percentages calculated based upon total number of patients with positive cultures (103); [b] Percentages calculated based upon total number of positive cultures (82) of patients diagnosed with clinical criteria for PJI.
(TIF)

**S2 Table. Cases of PJI diagnosed by the clinical modified criteria with discordant results between tissue culture and sonication fluid culture.** PJI: diagnosis of prosthetic joint infection according to clinical modified criteria; ATB: antimicrobial use in the 15 days leading up to prosthetic joint revision surgery.
(TIF)

**S3 Table. Cases of PJI diagnosed by the clinical modified criteria with discrepant results between tissue culture and sonication fluid culture.** PJI: diagnosis of prosthetic joint infection according to clinical criteria; ATB: antimicrobial use in the 15 days leading up to prosthetic joint revision surgery.
(TIF)

**S1 Dataset. Fully anonymized dataset.** Direct and indirect participant identifiers have been withdrawn from the dataset according to the instructions on preparing raw clinical data for publication provided by PLOS.
(XLSX)

**S1 Data. Metadata with values underlying reported findings.**
(XLSX)

**S2 Data. Metadata with values used to build tables and graphs.**
(XLSX)

## Acknowledgments

We thank Maria Aparecida Soares Murça (Department of Laboratory Medicine and Pathology- Irmandade Santa Casa de Misericórdia de São Paulo, São Paulo, Brazil) for providing key support in the sonication technique at the microbiology laboratory, and to Erika Tiemi Fukunaga (Research support center of the Faculdade de Ciências Médicas da Santa Casa de São Paulo; São Paulo, Brazil) who contributed in the statistical analysis of our study.

## Author Contributions

**Conceptualization:** Giselle Burlamaqui Klautau, Mauro Jose Salles.

**Data curation:** Taiana Cunha Ribeiro, Emerson Kiyoshi Honda, Daniel Daniachi, Ricardo de Paula Leite Cury.

**Formal analysis:** Taiana Cunha Ribeiro, Cely Barreto da Silva.

**Investigation:** Taiana Cunha Ribeiro, Cely Barreto da Silva, Giselle Burlamaqui Klautau.

**Methodology:** Mauro Jose Salles.

**Project administration:** Cely Barreto da Silva, Mauro Jose Salles.

**Resources:** Emerson Kiyoshi Honda, Daniel Daniachi, Ricardo de Paula Leite Cury, Giselle Burlamaqui Klautau.

**Supervision:** Mauro Jose Salles.

**Validation:** Taiana Cunha Ribeiro.

**Visualization:** Taiana Cunha Ribeiro.

**Writing – original draft:** Taiana Cunha Ribeiro.

**Writing – review & editing:** Mauro Jose Salles.

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
