## [Decision Letter · Decision Letter 0]

20 Apr 2021

PONE-D-21-04964

The impact of sonication cultures when the diagnosis of prosthetic joint infection is inconclusive

PLOS ONE

Dear Dr. Salles,

Thank you for submitting your manuscript to PLOS ONE. After careful consideration, we feel that it has merit but does not fully meet PLOS ONE’s publication criteria as it currently stands. Therefore, we invite you to submit a revised version of the manuscript that addresses the points raised during the review process.

Both reviewers noted that the manuscript is well-written. In your resubmission, please respond to the specific points and expand the discussion re: Reviewer's 2 suggestion.

We look forward to receiving your revised manuscript.

Kind regards,

Noreen J. Hickok, Ph.D.

Academic Editor

PLOS ONE

Journal Requirements:

Reviewers' comments:

Reviewer's Responses to Questions

**Comments to the Author**

1. Is the manuscript technically sound, and do the data support the conclusions?

Reviewer #1: Yes

Reviewer #2: Yes

2. Has the statistical analysis been performed appropriately and rigorously? 

Reviewer #1: Yes

Reviewer #2: Yes

3. Have the authors made all data underlying the findings in their manuscript fully available?

Reviewer #1: Yes

Reviewer #2: Yes

4. Is the manuscript presented in an intelligible fashion and written in standard English?

Reviewer #1: Yes

Reviewer #2: Yes

5. Review Comments to the Author

Reviewer #1: This is a well-written retrospective cohort study that aims to analyze the sensitivity and specificity of sonicate culture vs. standard tissue culture acquisition amongst individuals with TKA/THA PJI’s, through the comparison of a variety of clinical guidelines.

Some generalized points:

What cutoff is being used for definition of positive sonicate cultures? Greater than or equal to 5 CFU? This should be defined, if not already done, in the paper-maybe I missed it?

Only minor suggestions, as outlined below:

Lines 42-44: Did tissue cultures provide greater specificity or sensitivity? First part of sentence mentions specificity, but then sensitivity comparison is given (100% vs 95.3%)

Line 111: Changed backed-up to back-up

Line 134: Change ‘Institution’ to ‘institution’

In table 1: Capitalize the word ‘-mean’

Line 162: Remove the phrase ‘Worthy mentioning’

Line 271: ‘IDSA guideline’ should be ‘IDSA guidelines’

Line 293: Remove comma

Line 341: Should be ‘In routine clinical practice’

Line 342: Remove comma

Line 345: Change ‘independently’ to independent

Reviewer #2: This is a very well written clinical retrospective. Importantly, it corroborates research findings and observations widely published in a clinical hospital setting treating PJI patients. Specifically, the observation of low frequency ultrasound (U/S) and its associated cavitation nuclei behavior has been widely recognized in many laboratories in a research setting. I encourage the authors to briefly expand the discussion to the effect where U/S not only aids in removing biofilm from hardware but more importantly also acts as a biofilm dispersant. It is this phenomenon most likely responsible for increased microbiological identification in vitro.

6. PLOS authors have the option to publish the peer review history of their article (what does this mean?). If published, this will include your full peer review and any attached files.

Reviewer #1: No

Reviewer #2: No

---

## [Author Response · Author response to Decision Letter 0]

6 May 2021

MANUSCRIPT ID: PONE-D-21-04964

Article Title: The impact of sonication cultures when the diagnosis of prosthetic joint infection is inconclusive

Journal: PLOS ONE

On behalf of my coauthors, I am pleased to resubmit our amended manuscript to Plos One, and would like to extend our gratitude to the reviewers who have helped us improve our manuscript, by providing constructive and useful comments. We have revised our manuscript and have addressed the comments and queries provided by the reviewer below, point by point. Moreover, three new references (35-37) were added to the Reference list to back up the expansion of the discussion. The revised manuscript has also been submitted with as an unmarked and marked (with all changes tracked) version.

Thank you very much for the opportunity to resubmit our revised manuscript. 

Reviewer Comments:

Reviewer #1: This is a well-written retrospective cohort study that aims to analyze the sensitivity and specificity of sonicate culture vs. standard tissue culture acquisition amongst individuals with TKA/THA PJI’s, through the comparison of a variety of clinical guidelines. 

Authors’ response: We thank the reviewer for these thoughtful comments. 

Some generalized points:

What cutoff is being used for definition of positive sonicate cultures? Greater than or equal to 5 CFU? This should be defined, if not already done, in the paper-maybe I missed it?

Authors’ response: The authors thank the reviewer for addressing this important issue. Due to the previous experience of our research group with sonication fluid cultures on the diagnosis of prosthetic joint infection, we have published two studies (Yano et al., 2014; Zitron et al., 2016) that validated a cutoff of 50 CFU/plate. The reason for that particular cutoff number is because we added a concentration step of sonication fluid using 50-ml aliquots at 2,500 rpm for 5 min. Briefly, on this step the supernatant was aspirated leaving 0.5 ml (100-fold concentration), and aliquots of 0.1 ml of concentrated sonicate fluid were then plated onto agar plates and incubated. Due to the addition of a concentrating step in the process, a cutoff of 50 CFU/plate was considered positive and used for ideal sensitivity and specificity analyses. On the Material and Method Section, subitem Arthroplasty Sonication (Line 187- 218), the sonication fluid laboratory work up was described in detail from line 199 to line 211. Both articles are referenced in line 211.

Only minor suggestions, as outlined below:

Lines 42-44: Did tissue cultures provide greater specificity or sensitivity? First part of sentence mentions specificity, but then sensitivity comparison is given (100% vs 95.3%).

Authors’ response: Thank you for pointing out this typographical error, which we have now amended throughout the text. Indeed, according to Table 2 (Line 277), tissue culture provided greater specificity than sonication culture for all the criteria assessed, except for the EBJIS criteria, in which sonication than tissue cultures specificity was 100% and 95.3% (95% CI: 87.8-100%), respectively (p=0.024).

Line 111: Changed backed-up to back-up

Authors’ response: We thank the reviewer for pointing out this error, which we have now corrected in the text.

Line 134: Change ‘Institution’ to ‘institution’

Authors’ response: We thank the reviewer for pointing out this typographical error, which we have now corrected in the text.

In table 1: Capitalize the word ‘-mean’

Authors’ response: We thank the reviewer for pointing out this error, which we have now corrected in Table 1.

Line 162: Remove the phrase ‘Worthy mentioning’

Authors’ response: We amended the text, by deleting “(Worthy mentioning that)” on line 263.

Line 271: ‘IDSA guideline’ should be ‘IDSA guidelines’

Authors’ response: We thank the reviewer for pointing out this typographical error, which we have now corrected in the text.

Line 293: Remove comma

Authors’ response: We amended the text as requested. 

Line 341: Should be ‘In routine clinical practice’

Authors’ response: We amended the text as requested. 

Line 342: Remove comma

Authors’ response: We amended the text as requested. 

Line 345: Change ‘independently’ to independent

Authors’ response: We thank the reviewer for pointing out these typographical errors, which we have now corrected throughout the text.

Reviewer #2: This is a very well written clinical retrospective. Importantly, it corroborates research findings and observations widely published in a clinical hospital setting treating PJI patients. Specifically, the observation of low frequency ultrasound (U/S) and its associated cavitation nuclei behavior has been widely recognized in many laboratories in a research setting. I encourage the authors to briefly expand the discussion to the effect where U/S not only aids in removing biofilm from hardware but more importantly also acts as a biofilm dispersant. It is this phenomenon most likely responsible for increased microbiological identification in vitro.

Authors’ response: The reviewer addresses an important research question, and the authors do agree that the mechanisms through which sonication process successfully increases the diagnostic yield of microorganisms from biofilms, must be assessed in the Discussion Section. Indeed, the effects of sonication on cells or molecular aggregates applied at low frequency (40 ± 2 kHz) and high intensity of 0.22 ± 0.04 W/cm2, has been proved to increase the accuracy of bacterial counts within cultures by disrupting aggregates without loss of bacterial viability (Totten AH, et al. J Microb Met 2017). This effect is related to a simple mechanical destruction of the biofilm extracellular matrix due to the effect of ultrasonic cavitation (derived by High intensity and Low frequency US). Sonication enables detachment percentages of bacterial cells of 85–90% after US low frequency application (from 40 to 100 kHz) (Erriu M, et al. Ultrason Sonochem 2014). In addition, the natural process of biofilm dispersion, in which cell escape from the inner biofilm structure to return to its previous single-cell planktonic mode of growth, is likely increased by sonication process (Rumbaugh KP, et al. Nat Rev Microbiol 2020). Regardless the uncertainty of the current native mechanisms on how microbial disperses from biofilm, it is thought to occur actively in response to signaling molecules or cues that are synthesized by the resident biofilm cells. (Rumbaugh KP, et al. Nat Rev Microbiol 2020). Respond to environmental stimuli such as nutrients, nitric oxide, oxygen concentration or even changing conditions of the surrounding environment may play a role in the so called ‘environmentally induced dispersion’ of biofilm cells. Passive biofilm dispersion or biofilm detachment relies on external physical triggers that result in the release of single cells or clumps of biofilms (Wille J, et al. Biofilm 2020). Low-powered sonication has been proved to increase the dispersion of biofilm cells (Totten AH, et al. J Microb Met 2017). We expended the discussion by assessing the underlying mechanisms that makes sonication an effective tool for microbiological identification of implant-association biofilm infections (Line 407 to 414). Three new references (35-37) were added to the Reference list to back up this discussion.

Authors’ response: The authors thank the reviewer for the helpful assessment of the manuscript. We are convinced that answering the following questions improves the manuscript and makes it easier for the reader to understand and reflect the results in light of current research.

Thank you again for considering this manuscript for your journal.

Kind regards,

Mauro Jose Costa Salles, MD, MSc, PhD 

Assistant Professor

Infectious Diseases Clinic, Internal Medicine Department

Santa Casa de São Paulo School of Medical Sciences

Rua Dr Cesareo Mota Jr 112, CEP: 01221-020, São Paulo, SP, Brazil

Office: +55 11 21767706/21146262

Mobile: + 55 11 985360055; Fax: + 55 11 21146363

---

## [Editor Report · Decision Letter 1]

14 May 2021

The impact of sonication cultures when the diagnosis of prosthetic joint infection is inconclusive

PONE-D-21-04964R1

Dear Dr. Salles,

We’re pleased to inform you that your manuscript has been judged scientifically suitable for publication and will be formally accepted for publication once it meets all outstanding technical requirements.

Kind regards,

Noreen J. Hickok, Ph.D.

Academic Editor

PLOS ONE

---

## [Editor Report · Acceptance letter]

1 Jul 2021

PONE-D-21-04964R1 

The impact of sonication cultures when the diagnosis of prosthetic joint infection is inconclusive 

Dear Dr. Salles:

I'm pleased to inform you that your manuscript has been deemed suitable for publication in PLOS ONE. Congratulations! Your manuscript is now with our production department. 

Kind regards, 

on behalf of

Dr. Noreen J. Hickok 

Academic Editor

PLOS ONE